# High Alcohol Intake in Older Men and the Probability of Osteoporotic Fracture According to the FRAX Algorithm

**DOI:** 10.3390/nu13092955

**Published:** 2021-08-25

**Authors:** Julie A. Pasco, Kara B. Anderson, Natalie K. Hyde, Lana J. Williams, Pamela Rufus-Membere, Kara L. Holloway-Kew

**Affiliations:** 1IMPACT—Institute for Mental and Physical Health and Clinical Translation, Deakin University, Geelong, VIC 3220, Australia; kbanders@deakin.edu.au (K.B.A.); natalie.hyde@deakin.edu.au (N.K.H.); l.williams@deakin.edu.au (L.J.W.); pamela.r@deakin.edu.au (P.R.-M.); k.holloway@deakin.edu.au (K.L.H.-K.); 2Department of Medicine-Western Health, The University of Melbourne, St. Albans, VIC 3010, Australia; 3Department of Epidemiology and Preventive Medicine, Monash University, Melbourne, VIC 3004, Australia

**Keywords:** fracture risk, alcohol, bone fragility, injury, FRAX algorithm, major osteoporotic fracture

## Abstract

We aimed to determine the contribution of high alcohol intake to fracture probability, calculated using a fracture-risk assessment tool (FRAX). Participants were 262 men (ages 60–90 y) in the Geelong Osteoporosis Study. Alcohol consumption was documented via a food frequency questionnaire; 46 (17.6%) consumed three or more units per day, fulfilling the criterion for high alcohol intake. Bone mineral density (BMD) was measured by dual energy x-ray absorptiometry. We determined FRAX probabilities (%) for major osteoporotic fracture (MOF) and hip fracture (HF), calculated with and without alcohol intake. Thresholds for high FRAX probabilities, calculated with or without BMD, were ≥20% for MOF and ≥3% for HF. Proportions of men with high HF-FRAX probabilities were consistently greater for drinkers compared with non-drinkers. For drinkers, paired differences showed that median MOF-FRAX_withoutBMD_ probabilities calculated with and without alcohol changed by −2.3, HF-FRAX_withoutBMD_ by −1.7, MOF-FRAX_withBMD_ by −1.4, and HF-FRAX_withBMD_ by −0.9 (all *p* < 0.001). We estimated that, should drinkers lower their alcohol consumption to <3 units/d, up to 66.7% of those at high risk for MOF and up to 41.0% at high risk for HF would reduce their FRAX probabilities to below the thresholds for high fracture risk. In the context of the Australian environment, these data describe the extent to which older men with high alcohol consumption are at increased risk for fracture.

## 1. Introduction

The University of Sheffield in the United Kingdom developed the FRAX algorithm to estimate absolute fracture risk from the combination of several clinical risk factors [1]. The FRAX algorithm provides a calculated estimate for a 10-year probability of major osteoporotic fracture (MOF; including hip, spine, forearm, and proximal humerus fractures) and hip fracture (HF) by integrating up to eleven clinical risk factors, namely age, sex, weight, height, previous fracture, parental hip fracture, smoking, glucocorticoid use, rheumatoid arthritis, secondary osteoporosis, alcohol consumption, and bone mineral density (BMD) at the femoral neck. The inclusion of BMD is optional to enable FRAX probabilities to be calculated even if densitometric assessment is not possible. In the FRAX algorithm, high alcohol consumption is recognised as three or more units per day.

While some studies have reported that moderate alcohol consumption is associated with health benefits, including higher BMD and reduced risk for fracture [2,3,4], chronic excessive alcohol consumption has been associated with detrimental skeletal effects, including low BMD and elevated risk for fracture [5,6].

A report in 2018 from the World Health Organisation revealed that Australians rank among the heaviest alcohol consumers in the world [7]. Moderate to heavy alcohol consumption can displace dietary nutrients important for bone health [8] and is associated with accumulation of visceral fat [9], lower levels of vitamin D [10], and increased risk for falls and fall-related fractures [11,12]. Alcohol also directly affects bone metabolism by reducing the activity of both osteoblasts and osteoclasts [13], with reductions in bone formation and resorption occurring independent of calciotropic hormones [6]. We aimed to determine the contribution of high alcohol consumption to increased risk for MOF and HF for older men residing in Australia.

## 2. Materials and Methods

### 2.1. Participants

This cross-sectional analysis is part of a randomly recruited prospective cohort study of men enrolled in the Geelong Osteoporosis Study (GOS) [14]. The GOS is a population-based study designed to describe the health burden of osteoporosis and identify risk factors for fragility fracture. At baseline (2001–2006), an age-stratified sample of 1540 men aged 20–97 years was recruited from electoral rolls for the Barwon Statistical Division, with a participation of 67%. Five years later, 978 men were re-assessed at follow-up; reasons for loss to follow-up have been detailed elsewhere [14]. In this study, we utilised data from the 5-year re-assessment phase for men aged 60 years and over, as they are at increased risk for hip and other major osteoporotic fractures [15]. Two hundred and sixty-two men provided data for analysing FRAX probabilities and were thus included in analyses. All participants provided informed, written consent. Ethics approval was obtained from the Human Research Ethics Committee at Barwon Health.

### 2.2. Data

The Australian version of the FRAX (FRAX Aus^®^) included the following clinical risk factors: age, sex, weight, height, previous fracture, parental hip fracture, current smoking, glucocorticoid use, rheumatoid arthritis, secondary osteoporosis, alcohol consumption of ≥3 units/day, and BMD at the femoral neck. Data were entered into the FRAX (Aus) online tool for each participant (https://www.sheffield.ac.uk/FRAX/tool.aspx?country=31, accessed on 20 July 2021), and separate 10-year probability scores were generated for MOF (fractures of the hip, spine, wrist, and proximal humerus) and hip fractures (HF) and expressed as a percentage. FRAX scores were calculated with BMD (FRAX_withBMD_) and without BMD (FRAX_withoutBMD_).

BMD was measured at the femoral neck using dual-energy X-ray absorptiometry (DXA; GE Lunar, Prodigy Pro, Madison, WI, USA). Weight was measured to ±0.1 kg using electronic scales, and height was measured to ±0.001 m using a wall-mounted Harpenden stadiometer. Previous fractures were self-reported and verified from radiology reports where possible. Smoking, medication use, rheumatoid arthritis, and secondary osteoporosis were documented by questionnaire, as previously described [16,17,18]. Secondary osteoporosis included type 1 diabetes, osteogenesis imperfecta in adults, untreated long-standing hyperthyroidism, malabsorption, and chronic liver disease. Area-based socioeconomic status was ascertained using Socio-Economic Index for Areas index scores based on census data from the Australian Bureau of Statistics. These data were used to derive an Index of Relative Socio-Economic Advantage and Disadvantage (IRSAD) that was categorised into five groups according to quintiles of IRSAD for the study region.

Alcohol consumption was estimated using a food-frequency questionnaire from the Cancer Council Victoria [19]. The questionnaire captures the usual dietary habits during the preceding 12 months and encompasses five types of dietary intake, including alcoholic beverages, using a 10-point frequency scale; the usual number of glasses of beer, spirits, and/or wine consumed each day of drinking was also captured on a 10-point scale. In Australia, a standard drink contains 10 g of alcohol, and this corresponds to one glass of full-strength beer (285 mL), a single measure of spirits (30 mL), or a medium-sized glass of wine (120 mL). Aligning with the FRAX (Aus) guidelines, mean daily intakes ≥3 units of alcohol (equivalent to three or more a standard drinks) identified drinkers.

For drinkers, FRAX scores were calculated with and without a positive response to alcohol intake, and differences were compared to determine the impact on fracture risk if alcohol consumption were to be reduced to below 3 units per day. FRAX cut-points of ≥20% for MOF and ≥3% for hip fracture were adopted to identify those at high risk for fracture [20].

### 2.3. Statistics

Differences between alcohol drinkers (≥3 units/d) and non-drinkers (<3 units/d) were identified using *t*-tests or Mann–Whitney for continuous variables with a normal or skewed distribution, respectively, and the χ^2^ test for categorical variables (employing Fisher’s exact test when expected cell count <5 in 2 × 2 tables). Differences in median FRAX probabilities for age categories 60–69, 70–79, and 80+ years were tested using the Kruskal–Wallis test. For the drinkers only, differences in median FRAX probabilities calculated with and without alcohol were determined using Wilcoxon signed-rank test. Logistic regression was used to investigate the likelihood of high FRAX probabilities (MOF-FRAX > 20% and HF-FRAX > 3%) in association with age (categories 60–69, 70–79, and 80+ years) and alcohol drinkers (yes/no). Statistical analyses were performed using Minitab (v16, USA).

## 3. Results

### 3.1. All Participants

Participant characteristics are shown in Table 1. Alcohol intakes ranged from 0 to 95 g/d. Among 262 men, 46 (17.6%) had high alcohol intakes (≥3 units/d), fulfilling the criterion for an alcohol drinker in the FRAX algorithm. The numbers of men identified as drinkers were 19 (17.3%) for age category 60–69 y, 17 (18.2%) for 70–79 y, and 10 (17.0%) for 80+ y. Compared to non-drinkers, drinkers were more likely to have a parent with a history of hip fracture, and there was some evidence to suggest that drinkers were more likely to have had a previous fracture (*p* = 0.053). No differences in socioeconomic status were detected between men who consumed ≥3 or <3 units alcohol/d.

FRAX probabilities for MOF and HF with or without BMD were significantly greater for drinkers (Table 1, Figure 1). Figure 1 also shows that FRAX probabilities increased across age categories 60–69, 70–79, and 80+ years (all *p* < 0.001). The proportions of men with HF-FRAX probabilities ≥3% calculated with or without BMD were consistently greater for drinkers compared with non-drinkers.

In a multivariable model, the likelihood for high HF-FRAX_withBMD_ probability increased sequentially for age categories 60–69, 70–79, and 80+ years and was three-fold greater for alcohol drinkers compared to non-drinkers (Table 2). Although patterns of high alcohol intakes and increasing age categories were observed for HF-FRAX_withoutBMD_ and MOF-FRAX with or without BMD (Figure 1), numbers of participants in subgroups were too small for meaningful analyses using these algorithms for fracture risk.

### 3.2. Alcohol Drinkers

For the drinkers, re-calculation of FRAX probabilities without the contribution from alcohol reduced median FRAX probabilities. Median MOF-FRAX_withoutBMD_ probabilities changed from 9.1% to 6.8%, HF-FRAX_withoutBMD_ from 4.9% to 3.2%, MOF-FRAX_withBMD_ from 5.5% to 4.3%, and HF-FRAX_withBMD_ from 1.9% to 1.3%. Paired differences showed that median MOF-FRAX_withoutBMD_ probabilities calculated with and without alcohol changed by −2.3 (W = 1081, *p* < 0.001), HF-FRAX_withoutBMD_ by −1.7 (W = 1081, *p* < 0.001), MOF-FRAX_withBMD_ by −1.4 (W = 1081, *p* < 0.001), and HF-FRAX_withBMD_ changed by −0.9 (W = 946, *p* < 0.001).

If alcohol drinkers were to lower alcohol intake to <3 units/d and thus lower their FRAX probabilities, two of three drinkers (66.7%) would have MOF-FRAX_withoutBMD_ probabilities below the threshold for high risk for MOF; similarly, three of 28 drinkers (10.7%) would lower their HF-FRAX_withoutBMD_ probabilities, and seven of 17 drinkers (41.2%) would lower their HF-FRAX_withBMD_ probabilities below the threshold for high risk for HF. The sole drinker with high MOF-FRAX_withBMD_ probability would remain at high risk for MOF even if they became a non-drinker.

## 4. Discussion

Here, we report that men who consumed three or more units of alcohol per day were at greater risk for MOF and HF than peers who consumed less than three units per day and that increased fracture risk was sequentially more pronounced for older age categories. We estimated that should drinkers lower their alcohol consumption to below three units per day, up to 66.7% of those at high risk for MOF and up to 41.0% at high risk for HF would reduce their risk to below the thresholds of 20% for MOF and 3% for HF.

Alcohol is considered a non-essential component of diet. The association between alcohol consumption and bone health is not well understood because of the interplay between the actions of alcohol on bone metabolism other alcohol-related chronic health condition and the co-occurrence of heavy alcohol consumption with a range of other poor lifestyle choices.

A systematic review and meta-analysis conducted in 2008 revealed that, in comparison with alcohol abstinence, daily consumption of one drink or less was associated with a lower risk of hip fracture, whereas more than two drinks was associated with higher hip fracture risk [4]. They also reported that alcohol consumption of up to two drinks per day was linearly associated with greater BMD. A more recent systematic review and meta-analysis revealed that, compared to alcohol abstainers, drinkers were at greater risk for osteoporosis; the authors reported a dose-effect whereby daily consumption of 0.5–1 drinks was associated with 0.90–2.12-fold increased risk, 1–2 drinks with 1.11–1.62-fold increased risk, and two drinks or more with 1.01–2.65 times the risk [21].

Our data revealed that for alcohol drinkers, differences between FRAX probabilities calculated with and without alcohol were greater for algorithms that did not include BMD. This supports the notion that extra-skeletal sequelae of high alcohol consumption, such as increased risk for falls and fall-related fractures [11,12], contribute to fracture risk independent of alcohol-related deficits in BMD [22]. It should also be noted that there are other constituents of alcoholic beverages that can differentially affect bone metabolism, such as silicon in beer, which has a positive effect on BMD [23]; the FRAX algorithm does not account for the type of alcoholic beverage consumed.

A major strength of our study is that participants were drawn from the broad population and were unselected in terms of health behaviours and disease status. Where possible, we used objective measures to calculate FRAX probabilities. However, we acknowledge that we relied on some self-reported data, particularly alcohol consumption, which may be subject to differential recall bias according to the quantities of alcohol consumed. We utilised a food-frequency questionnaire designed in Australia for use in epidemiological studies to determine alcohol consumption [19]. In order to compare differences in FRAX probabilities calculated with and without consideration of high alcohol consumption, we did not account for potential differences in other clinical risk factors that might correspond to changes in alcohol consumption. Further, the lifestyle risk factors included in FRAX are likely to have a dose effect on the risk for fracture; however, we complied with the recommended threshold of three or more units per day as indicative of high alcohol intake when utilising the FRAX algorithm to estimate fracture probabilities. The sample size in this study was small, so results should be interpreted with caution; moreover, small numbers limited our ability to use statistical modelling to simultaneously identify the comparative contributions to fracture probability of all clinical risk factors included in the FRAX algorithm. Data were collected for men residing in southeastern Australia, and we recognise that the results may not be generalisable to women nor to other populations of men. As we used cross-sectional data, causality cannot be inferred. Intervention studies are needed to confirm our results. 

## 5. Conclusions

We used a theoretical model to quantify the extent to which men with high alcohol consumption are at increased risk for fracture, and we identified the proportion of drinkers categorised as being at high risk for MOF and HF that could reduce their fracture risk if they lowered their alcohol consumption to below three units per day.

## Figures and Tables

**Figure 1 nutrients-13-02955-f001:**
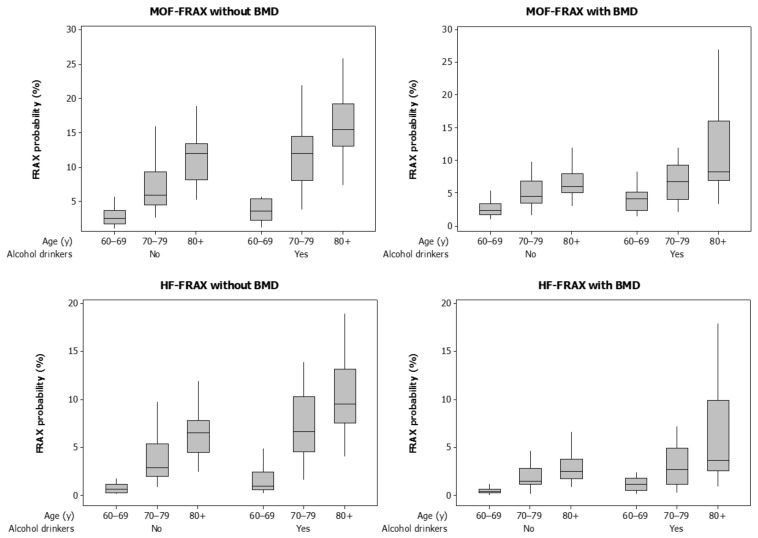
Boxplots for FRAX probabilities calculated with and without bone mineral density (BMD) for major osteoporotic fractures (MOF) and hip fractures (HF). Data are shown for alcohol drinkers (alcohol ≥ 3 units/d) and non-drinkers (alcohol < 3 units/d) and by age category.

**Table 1 nutrients-13-02955-t001:** Participant characteristics for the whole group and according to alcohol consumption (drinkers ≥ 3 and non-drinkers < 3 units alcohol/d). Data are shown as mean (±SD), median (interquartile range), or *n* (%).

	All	Drinkers		
	*n* = 262	Yes (*n* = 46)	No (*n* = 216)	*p*
Age (y)	72.4 (65.8–79.3)	72.2 (63.8–78.7)	72.4 (65.9–79.5)	0.588
Alcohol (g/d)	11 (1–25)	45 (36–61)	7 (0–17)	<0.001
Weight (kg)	82.9 (±13.3)	83.3 (±14.0)	82.8 (±13.2)	0.846
Height (m)	1.73 (±0.07)	1.74 (±0.08)	1.73 (±0.06)	0.381
BMI (kg/m^2^)	27.7 (±4.0)	27.6 (±4.3)	27.7 (±4.0)	0.800
BMD * (g/cm^2^)	0.943 (±0.134)	0.939 (±0.155)	0.944 (±0.129)	0.856
Previous fracture	109 (41.6%)	25 (54.4%)	84 (38.9%)	0.053
Parental hip fracture	23 (8.8%)	8 (17.4%)	15 (6.9%)	0.039
Current smoker	14 (5.3%)	4 (8.7%)	10 (4.6%)	0.278
Glucocorticoid user	16 (6.1%)	4 (8.7%)	12 (5.6%)	0.494
Rheumatoid arthritis	10 (3.8%)	1 (2.2%)	9 (4.2%)	0.999
Secondary osteoporosis **	33 (12.6%)	5 (10.9%)	28 (13.0%)	0.698
Socioeconomic status ***				0.520
Quintile 1	44 (16.8)	6 (13.0%)	38 (17.6%)	
Quintile 2	58 (22.1%)	14 (30.4%)	44 (20.4%)	
Quintile 3	53 (20.2%)	7 (15.2%)	46 (21.3%)	
Quintile 4	55 (21.0)	11 (23.9%)	44 (20.4%)	
Quintile 5	52 (19.9%)	8 (17.4%)	44 (20.4%)	
MOF-FRAX_withoutBMD_	5.3 (3.0–10.0)	9.1 (3.8–15.0)	5.1 (2.8–9.0)	0.001
HF-FRAX_withoutBMD_	2.4 (0.9–5.6)	4.9 (1.2–8.7)	2.0 (0.8–5.0)	0.002
MOF-FRAX_withBMD_	4.1 (2.7–6.5)	5.5 (3.6–9.1)	3.8 (2.5–5.8)	0.001
HF-FRAX_withBMD_	1.3 (0.5–2.6)	1.9 (0.9–4.3)	1.2 (0.5–2.2)	0.002
MOF-FRAX_withoutBMD_ ≥ 20%	7 (2.7%)	3 (6.5%)	4 (1.9%)	0.106
HF-FRAX_withoutBMD_ ≥ 3%	113 (43.1%)	28 (60.9%)	85 (39.4%)	0.007
MOF-FRAX_withBMD_ ≥ 20%	2 (0.8%)	1 (2.2%)	1 (0.5%)	0.321
HF-FRAX_withBMD_ ≥ 3%	55 (21.0%)	17 (37.0%)	38 (17.6%)	0.003

Abbreviations: BMI, body mass index; BMD, bone mineral density; MOF, major osteoporotic fracture; HF, hip fracture. * BMD at the femoral neck. ** Includes type 1 diabetes, osteogenesis imperfecta in adults, untreated long-standing hyperthyroidism, malabsorption, and chronic liver disease. *** Quintile 1 is the most disadvantaged, and quintile 5 is the least disadvantaged.

**Table 2 nutrients-13-02955-t002:** Logistic regression model for high HF-FRAX calculated with BMD (probabilities ≥ 3%) in association with age category and alcohol consumption.

		OR (95%CI)	*p*
Age category	60–69 y	Reference	-
	70–79 y	11.1 (3.6, 33.7)	<0.001
	80+ y	22.2 (7.0, 70.1)	<0.001
Alcohol consumption	<3 units/d	Reference	-
	>3 units/d	3.48 (1.56, 7.77)	0.002

## Data Availability

The data that support the findings of this study are available from the corresponding author upon reasonable request.

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
