# Peer review of "High Alcohol Intake in Older Men and the Probability of Osteoporotic Fracture According to the FRAX Algorithm"

_nutrients, 2021, doi:10.3390/nu13092955_

Round 1

Reviewer 1 Report

The revised manuscript  showed that  that men who consumed three or more units of alcohol per day were at greater risk for major osteoporotic fracture (MOF) and hip fracture (HF), than peers who consumed less than three units per day, and that increased fracture risk was sequentially more pronounced for older age categories. In my opinion this manuscript is  interesting but  suffers from some significant  flaws.

  1. The title of this manuscript and key words should include the information that the FRAX algorithm was used.
  2. Please indicate the novelty of this work. There are many works in the past that show the association between alcohol consumption and osteoporotic fracture. For that reason the alcohol intake was included in the FRAX calculator.
  3. Please include the address of a web page for FRAX calculator that you used (line76).
  4. Please include information how the amount of alcohol in grams [Alcohol (g/d)] was calculated.
  5. The logistic regression was calculated for the two components of the FRAX algorithm (age and alcohol consumption). It will be good to compare all individual components of the FRAX algorithm in logistic analysis. This analysis will allow to compare the strength of the variables included in the FRAX calculator.
  6. Socioeconomic status or nutritional deficiency are associated with a higher alcohol intake and probability of osteoporotic fracture, so these information could be included in your statistical analysis.
  7. The authors should include the limitation section, because this study has numerous limitations e.g. small sample size, self-reported data (data on alcohol consumption by self-report using simple surveys may lead to underreporting, particularly among heavy drinkers.
  8. Re-calculation of FRAX probabilities without the contribution from alcohol for the drinkers is not a good method for the estimation of risk reduction because a higher alcohol intake is associated with other factors that increased risk of osteoporotic fracture. The limitations of this study cause that these results must be interpreted with caution.

Author Response

Response to reviewers

High alcohol intake in older men and the probability of osteoporotic fracture according to the FRAX algorithm

Julie A Pasco, Kara B Anderson, Natalie K Hyde, Lana J Williams, Pamela Rufus-Membere and Kara L Holloway-Kew

Reviewer 1

The revised manuscript  showed that  that men who consumed three or more units of alcohol per day were at greater risk for major osteoporotic fracture (MOF) and hip fracture (HF), than peers who consumed less than three units per day, and that increased fracture risk was sequentially more pronounced for older age categories. In my opinion this manuscript is  interesting but  suffers from some significant  flaws.

  1. The title of this manuscript and key words should include the information that the FRAX algorithm was used.

Response: The title has been changed and now reads “High alcohol intake in older men and the probability of osteoporotic fracture according to the FRAX algorithm”. FRAX algorithm has been included in the list of key words.

  1. Please indicate the novelty of this work. There are many works in the past that show the association between alcohol consumption and osteoporotic fracture. For that reason the alcohol intake was included in the FRAX calculator.

Response: The FRAX algorithm includes several clinical risk factors, including alcohol (if consumption >3 unit/d), and FRAX probabilities can be calculated with and without BMD at the femoral neck. When BMD is included, it dominates the value of the FRAX score, and in the absence of BMD, the other clinical risk factors gain relative importance. Further, FRAX algorithms have been adjusted for each country, according to hip fracture rates. The coefficient of each factor and the contribution each risk factor makes to the overall FRAX probability is unknown. Details of the FRAX algorithm are not available in the literature. In this study, we have utilised real-life data to determine differences in FRAX probabilities calculated with and without high alcohol consumption, in models with and without BMD. Moreover, we determined the proportion of drinkers who would reduce their FRAX probabilities to below the thresholds for being at high risk of HF and MOF, should they reduce their alcohol intakes to < 3 units/d. It is not possible to calculate this information without these real-life data.

  1. Please include the address of a web page for FRAX calculator that you used (line76).

Response: The link has been included in Materials and Methods, section 2.2 (line 82).

  1. Please include information how the amount of alcohol in grams [Alcohol (g/d)] was calculated.

Response: The text has been amended to provide this information.

Section 2.2 (lines 99-106): The questionnaire captures the usual dietary habits during the preceding 12 months and encompasses five types of dietary intake, including alcoholic beverages, using a 10-point frequency scale; the usual number of glasses of beer, spirits and/or wine consumed each day of drinking was also captured on a 10-point scale. In Australia, a standard drink contains 10g of alcohol and this corresponds to one glass of full-strength beer (285 mL), a single measure of spirits (30 mL) or a medium-sized glass of wine (120 mL). Aligning with the FRAX (Aus) guidelines, mean daily intakes ≥3 units of alcohol (equivalent to three or more a standard drinks) identified drinkers.

  1. The logistic regression was calculated for the two components of the FRAX algorithm (age and alcohol consumption). It will be good to compare all individual components of the FRAX algorithm in logistic analysis. This analysis will allow to compare the strength of the variables included in the FRAX calculator.

Response: While we acknowledge that this is a good suggestion, our sample size is too small to simultaneously accommodate 11 exposure variables in the logistic regression model. We have now acknowledged this limitation in the text.

Section 4 (lines 216-219): The sample size in this study was small, so results should be interpreted with caution; moreover, small numbers limited our ability to use statistical modelling to simultaneously identify the comparative contributions to fracture probability of all clinical risk factors included in the FRAX algorithm.

  1. Socioeconomic status or nutritional deficiency are associated with a higher alcohol intake and probability of osteoporotic fracture, so these information could be included in your statistical analysis.

Response: We thank the reviewer for this suggestion. Socioeconomic status has now been included in the manuscript. We utilised an area-based index known as the Index of Relative Socio-Economic Advantage and Disadvantage (IRSAD); the IRSAD scores were categorised into quintiles. No differences in socioeconomic status were detected between men who consumed >3 or <3 units alcohol/day.

The text has been amended by adding the following:

Materials and Methods, section 2.2 (lines 93-97)): Area-based socioeconomic status was ascertained using Socio-Economic Index For Areas index scores based on census data from the Australian Bureau of Statistics. These data were used to derive an Index of Relative Socio-Economic Advantage and Disadvantage (IRSAD) that was categorised into five groups, according to quintiles of IRSAD for the study region.

Results, section 3.1 (lines 132-133): No differences in socioeconomic status were detected between men who consumed >3 or <3 units alcohol/d.

Table 1: Socioeconomic status has been added to the list of aggregate participant characteristics.

  1. The authors should include the limitation section, because this study has numerous limitations e.g. small sample size, self-reported data (data on alcohol consumption by self-report using simple surveys may lead to underreporting, particularly among heavy drinkers.

Response: We have added to the limitations section in the Discussion to address this comment.

Section 4 (lines 204-208): Where possible we used objective measures to calculate FRAX probabilities. However, we acknowledge that we relied on some self-reported data, particularly alcohol consumption, which may be subject to differential recall bias according to the quantities of alcohol consumed.

Section 4 (lines 216-219): The sample size in this study was small, so results should be interpreted with caution; moreover, small numbers limited our ability to use statistical modelling to simultaneously identify the comparative contributions to fracture probability of all clinical risk factors included in the FRAX algorithm.

  1. Re-calculation of FRAX probabilities without the contribution from alcohol for the drinkers is not a good method for the estimation of risk reduction because a higher alcohol intake is associated with other factors that increased risk of osteoporotic fracture. The limitations of this study cause that these results must be interpreted with caution.

Response: We acknowledge this limitation, as noted in lines 209-212 “In order to compare differences in FRAX probabilities calculated with and without consideration of high alcohol consumption, we have not accounted for potential differences in other clinical risk factors that might correspond to changes in alcohol consumption”.

Reviewer 2 Report

This manuscript is excellently written and was easy and interesting to read. The fracture probability would be of interest to readers, especially comparing fracture probability including and removing high alcohol intake. Research into fracture risk among men is lacking. Although values are in the tables, the authors may want to indicate significant differences and include other p values in the text.

Author Response

Response to reviewers

High alcohol intake in older men and the probability of osteoporotic fracture according to the FRAX algorithm

Julie A Pasco, Kara B Anderson, Natalie K Hyde, Lana J Williams, Pamela Rufus-Membere and Kara L Holloway-Kew

Reviewer 2

This manuscript is excellently written and was easy and interesting to read. The fracture probability would be of interest to readers, especially comparing fracture probability including and removing high alcohol intake. Research into fracture risk among men is lacking. Although values are in the tables, the authors may want to indicate significant differences and include other p values in the text.

Response: The minor word changes suggested by the reviewer in the annotated manuscript have been implemented. The changes are detailed below.

Section 2.1 (line 72): assessment has been changed to re-assessment

Section 3.1 (line 132): the p-value has been inserted.

Section 3.1 (line 142): the word ‘significantly’ has been inserted.

Round 2

Reviewer 1 Report

Thank you for responding to my comments. I have no additional comments.